

# Heterotic grouping of wheat hybrids based on general and specific combining ability from line × tester analysis

Birol Deviren[1], Oguz Bilgin[1] and Imren Kutlu[2]

[1] Field Crops Department, Agricultural Faculty, Namık Kemal University, Tekirdağ, Turkey
[2] Field Crops Department, Agricultural Faculty, Osmangazi University, Eskişehir, Turkey

## ABSTRACT

The most important step in plant breeding is the correct selection of parents, and it would be wise to use heterotic groups for this. The purpose of this study is to analyse yield and its components as well as genetic diversity in line × tester wheat populations. It also seeks to present a coherent framework for the isolation of early superior families and the development of heterotic groups in bread wheat. $F_1$ and $F_2$ generations of 51 genotypes, including 36 combinations between 12 lines and three testers and 15 parents, were evaluated for yield and its components in a three-replication experiment according to the randomized block design. Line × tester analysis of variance, general and specific combining abilities, heterosis, heterobeltiosis and inbreeding depression were calculated. Heterotic groups created based on general and specific combining abilities were compared with each other. The results showed that there was sufficient genetic variation in the population and that further genetic calculations could be made. The selections made based on general and specific combining abilities, heterosis values and average performance of genotypes without heterotic grouping indicated different genotypes for each feature. The creation of heterotic groups made it possible to select genotypes that were superior in terms of all the criteria listed. It was concluded that heterotic groups created based on specific combining abilities may be more useful for breeding studies.

Corresponding author
Imren Kutlu, ikutlu@ogu.edu.tr

## INTRODUCTION

Wheat is a crop that has an important role in human and animal nutrition all around the world. Studies continue globally to increase yield and quality of wheat. Because the projections made due to the continued increases in the world population predict that global food demands should double by 2050 (*FAOSTAT, 2024*). Reasons such as limited arable areas and increased production inputs emphasize that the increase in wheat production should largely result from unit area yield increases and innovations in plant breeding.

The success of any breeding program depends primarily on the proper selection of parents, the mating system to be used and finally the keen judgment of the breeder in selecting the desired superior genotypes within segregated populations. In this context,
it will make a significant contribution to the interpretation of genetic parameters by examining heterosis and inbreeding depression in most crops, including wheat. In the process of developing suitable breeding protocols, plant breeders can greatly benefit from an understanding of the kind and severity of heterosis and inbreeding depression (*Kuckuck, Kobabe & Wenzel, 1991*).

Estimation of genetic diversity based on genetic distance is useful for wheat breeding as a means of parental selection to promote new genetic recombination to increase grain yield. Crossbreeding followed by selection is one of the important methods of wheat breeding, and parent selection is the first step in a plant breeding program by crossing (*Khodadadi, Fotokian & Miransari, 2011*). Genetic distance between parents is essential to benefit from transgressive segregation. With greater genetic distance between parents, higher heterosis can be observed in the resulting offspring. Genetic distance estimation is one of the appropriate tools for parent selection in crossbreeding programs in wheat, and appropriate parent selection is required for use in crossing nurseries to improve genetic recombination to increase grain yield (*Dimitrov, Uhr & Velcheva, 2021*).

Appropriate identification of superior hybrid combinations in the $F_1$ generation is one of the most important issue in wheat breeding, as it will facilitate the identification of transgressive segregations that will occur in segregated generations. The phenotypic selection of promising parents can be performed based on either general combining ability (GCA) or line performance per se (*Adhikari et al., 2020*). Well-chosen parents produce an $F_1$ line of high heterotic potency that allows for the emergence of selectable transgressive variants and allows for the emergence of selectable transgressive variants in discriminating progeny. Unlike heterosis, transgressive phenotypes that occur as a result of transgressive segregation can be fixed after the $F_2$ generation. Although transgressive segregation in the $F_2$ generation has been predicted to result from heterozygous genotypes, as is the emergence of heterosis in the $F_1$ generation, there is little knowledge about the genetic basis of transgressive segregation (*Guindon et al., 2019*).

A key factor for the long-term success of wheat breeding understanding the aforementioned factors is the establishment of heterotic groups and the identification of a high-yielding heterotic model (*Boeven, Longin & Würschum, 2016*). Heterotic groups can be formed by crossbreeding, or pre-existing genetic variation can be used to define heterotic groups. *Melchinger & Gumber (1998)* proposed a multi-step procedure for the identification of heterotic groups, starting with genetic similarity-based germplasm grouping, followed by generation and evaluation of testcrosses, followed by a selection of promising cross combinations.

A heterotic group refers to a cluster of genotypes that, when crossed with another genetically different genotype group, exhibits comparable hybrid performance, while a heterotic model describes a particular pair of heterotic groups that show an optimal use of heterosis (*Melchinger & Gumber, 1998*). Heterosis can be regarded as a function of heterozygosity, and as a result, genetic variation among heterotic groups increases the level of heterosis (*Melchinger, 1999*; *Begna, 2021*). In genetically diverse heterotic populations, the ratio of variations attributable to specific combining ability (SCA) and general combining ability (GCA) is frequently low (*Gowda et al., 2010*). In other words, a
low intergroup ratio of $\sigma 2$SCA: $\sigma 2$GCA occurs relative to within-group crosses, indicating that the concept of heterotic patterns effectively supports the selection of superior hybrids. Furthermore, the dominance of GCA has also been shown to be beneficial for high genetic gains in recurrent selection, as well as for convenience in identifying promising hybrids based on GCA prediction (*Gupta et al., 2019*).

In particular, the magnitude and proportion of variance due to GCA and SCA are important for estimating the expected selection gain in hybrid breeding. Further, detailed knowledge of the hybrid performance, transgressive phenotypes, GCA and SCA effects, and heterotic groups based on them is required to optimize crossbreeding plans. For wheat, very limited information is available in the literature about these important quantitative genetic parameters (*Longin et al., 2013*; *Boeven, Longin & Würschum, 2016*; *Kutlu & Sirel, 2019*; *Shamsabadi et al., 2021*; *Argirou, 2023*). Therefore, it is of interest to examine the suitability of GCA- and SCA-based heterotic groups and to illustrate the usefulness of the approach for successful wheat breeding that selected hidden genetic entities producing extreme genotypes. This study aims to propose a unified framework for isolating early superior families and establishment of heterotic groups in bread wheat and to evaluate gene actions of yield and its components as well as genetic diversity in line × tester wheat populations. We highlighted the possibilities of using SCA-based heterotic groups in plant breeding in our previous study (*Kutlu & Sirel, 2019*). The most important unique value of our study, which distinguishes it from other similar studies, is that plant materials of different breeding origins were used in hybridizations (for example, anther culture mediated doubled-haploid, intergeneric cross line, gamma-induced mutant line), and it is interpreted by taking into account combining abilities (GCA and SCA), heterotic performances and inbreeding depression. Moreover, determining the effective parameters (GCA or SCA based) in creating heterotic groups and explaining the conveniences that heterotic group selection will offer in plant breeding make this study unique.

## MATERIALS & METHODS

### Parental genotype and crossing

The primary material was 15 diverse wheat genotypes that were taken and divided into two groups *viz.*, 12 females (lines) and three males (testers). Twelve females and three males were sown during 2014–2015 for crossing purposes following Line × Tester fashion (*Kempthorne, 1957*), at the experimental field of Tekirdağ Namık Kemal University, Faculty of Agriculture, Field Crops Department. All the females were crossed with three males to produce a sufficient $F_1$ seed of 36 crosses. The details of the genotypes are in Table 1.

### Experimental site and agronomic practices

The study was carried out to evaluate parents, $F_1$ and their $F_2$ for yield and some yield components during 2015–2016 and 2016–2017. Experiments set up with 51 genotypes including 15 parents and 36 crosses on 2 November 2015 ($F_1$) and on 10 November 2016 ($F_2$) in a randomized complete block design (RCBD) in three replications at the experimental field of Tekirdağ Namık Kemal University, Faculty of Agriculture, Field Crops Department. Each parent and $F_1$ were planted in a single row while each $F_2$ were

**Table 1** The pedigree and origins of lines and testers used in the study.

| Genotypes | Pedigree | Originate |
|---|---|---|
| NZFE 62 | KateA-I/Presto200 | Intergeneric cross line |
| NZFE 63 | Flamura85/Tatlıcak97 | Intergeneric cross line |
| NZFE 64 | Krasunia/Tatlıcak97 | Intergeneric cross line |
| NZFE 25 | Selianka/Syrena | Single cross line |
| NZFE 38 | Syrena/Flamura85 | Single cross line |
| NZFE 55 | Albatros/Victoria | Single cross line |
| NZFMT 14 | IBWSN4-100 | Gamma-induced mutant line |
| NZFMT 15 | IBWSN4-300 | Gamma-induced mutant line |
| NZFMT 21 | BEZOSTAYA-300 | Gamma-induced mutant line |
| 4166-1 | Sadovo/Sana | Anther culture mediated doubled-haploid line |
| 4162-28 | Krasunia Odeski/Sana | Anther culture mediated doubled-haploid line |
| 4164-36 | Golia/Sana | Anther culture mediated doubled-haploid line |
| Esperia | B 16/3*LINEA RUSSA | Italia |
| Tekirdağ | 1518-4-38K | Turkey |
| Renan | Mironovskaia 808/Maris Hunstman//VPM Moisson 4/Courtot | France |

planted in two rows of 2-meter-long plots and 20 cm apart. The plant-to-plant distance was maintained at five cm. Analysis results of the soil showed it had a clay-loam texture with 6.25 pH and showed weak soil characteristics. Considering recommended cultural practices, all the fertilizer doses of nitrogen 180 kg h$^{-1}$ and phosphorus 60 kg h$^{-1}$ were applied and other agronomic practices such as disease protection and weed control were achieved quickly.

Grain yield (GY) and some yield components such as plant height (PH), spike length (SL), number of grains per spike (GNS), grain weight per spike (GWS), thousand grain weight (TGW) and harvest index (HI) were investigated.

## Statistical analysis

The variance of GCA and SCA, heterosis (Ht), heterobeltiosis (Hb) and inbreeding depression (ID) were calculated as per standard procedures (*Fonseca & Patterson, 1968*; *Panse & Sukhatme, 1985*; *Kutlu & Sirel, 2019*). A cluster analysis was performed to set genotypes into groups by response type, based on parental means and GCA effects for the traits studied. Similarly, hybrids obtained from line × tester mating design were also clustered, starting with the construction of the degree of Euclidean distance matrix between genotypes and using the mean values of the hybrids and the effects of SCA as formerly described in *Kutlu & Sirel (2019)*. However, this study represents the investigation of GCA-based heterotic grouping, which was subjected to a comparative analysis with SCA-based grouping. Dendrograms were created according to the UPGMA method and heterotic groups were performed using the method suggested by *Anderberg (1993)* in the IBM SPSS 20 package program. In addition, a bi-plot graph was obtained by principal components analysis (PCA) to select the best parent/combination and the most influential factor for heterotic groups.

Genotype data for all traits examined were compared according to the mean values, GCA, SCA, Ht, Hb and ID values within heterotic groups formed according to cluster analysis. Heterosis, heterobelthiosis and inbreeding depression values of the parents were taken as the mean of the crossbreeding series in which they were included. It was not included since the parents in the crossbreed series had a mean of 0 for SCA because it only refers to the ability of the cross.

## RESULTS AND DISCUSSION

### Line ×tester variance analyses and estimations of combining abilities

The variations of all genotypes, parents and crosses were significant for all of the traits, except grain weight per spike in $F_1$ and harvest index in $F_2$ for parents (Table S1). This variation is evidence of the sufficient genetic variability between the genotypes for advanced genetic statistical evaluations.

Parent and hybrid interaction were significant for all traits except spike length and grain weight per spike in $F_1$, plant height, spike length and harvest index in $F_2$, and this indicates that the performance of the hybrids differed from their parents. The significant variation of lines and testers by trait or generation suggests that the contribution of lines or testers to the GCA variance components is different. The majority of the characteristics revealed that the "line × tester" interaction resulted in significant contributions to the SCA variance components, with hybrids playing a notable role.

Considering the mean performance of the parents, it was seen that although it differed for each trait, some of them showed higher values in the first year ($F_1$) than in the second year ($F_2$) of the research, while some showed the opposite (Table S2). These findings indicate that the examined traits are under the influence of the environment and the existence of genotype × environment interaction. The most outstanding parents were different for each trait and in each generation. In the $F_1$ generation, Renan for PH (91.7 cm), NZFE-63 for SL (17.0 cm), Tekirdağ for GNS (67.1 number), NZFE-62 for GWS (2.31 g) and HI (42.28%), Esperia for TGW (35.33 g), and NZFE-38 for GY (809 kg da$^{-1}$) were the most performing parents. While in the $F_2$ generation, NZFE-62 for PH (101.6 cm), Renan for SL (12.5 cm), Esperia for GNS (69.6 number), Tekirdağ for GWS (3.13 g), NZFMT-21 for TGW (48.25 g), 4162-28 for HI (52.51%), and NZFE-64 for GY (672 kg da$^{-1}$) were the remarkable genotypes. This made it difficult to select the parent with the highest yield and the best performance for yield components. Therefore, it would be more logical to choose according to the GCA of the parents for the traits we want to develop.

The negative GCA effect value emphasizes the reducing effect of the parent for the examined trait, while the positive GCA effect refers to the increasing contribution (*Fasahat et al., 2016*). The GCA values of parents are also different in each trait and generation, similar to their mean values (Table S3). Parents with positive GCA values for all traits except plant height should be selected. In terms of plant height, when we want to develop tall varieties, we should prefer NZFE-62 (8.86 **), NZFE-25 (6.04 **), NZFE-38 (7.02 **), NZFMT-14 (2.76 *), NZFMT-15 (6.70 **) and Renan genotypes as parents.

The performances of the combinations were also highly variable across traits and generations, similar to their parents (Table S4). Top performing cross combinations were in the $F_1$ generation, "NZFE-62/Renan" for PH (97.2 cm), "NZFE-63/Renan" for SL (13.7 cm), "NZFE-63/Tekirdağ" for GNS (69.7 number), "NZFMT-15/Tekirdağ" for GWS (3.25 g), "4164-36/Renan" for TGW (44.65 g), "NZFE-62/Esperia" for HI (45.37%), and "4162-28/Tekirdağ" for GY (847 kg da$^{-1}$). In the $F_2$ generation, "NZFE-38/Tekirdağ" for PH (98.7 cm), "4162-28/Renan" for SL (11.8 cm), "NZFE-63/Esperia" for GNS (74.4 number), "NZFE-25/Esperia" for GWS (3.54 g), "4166-1/Renan" for TGW (48.64 g), "4164-36/Tekirdağ" for HI (53.6%), and "NZFE-63/Renan" for GY (728 kg da$^{-1}$) were the highest value combinations. While the values of the cross combinations were higher in the $F_1$ generation for some traits, the values of some of them were higher in the $F_2$ generation. The reason why combinations show variable performance by trait may be due to the effect of genotype × environment interaction, as in the parents, or as an indication of high heterosis in $F_1$ or transgressive segregations in $F_2$.

The SCA effects are generally considered a good selection source for cross-pollinating species, however, it can also be used effectively to select homozygous lines with transgressive segregation (*Tiwari et al., 2023*). Positive and significant ones are also taken into consideration when choosing according to the SCA and it has been reported by many researchers that combinations with high value are promising in improving the relevant trait (*Kutlu & Sirel, 2019*; *Hakeem et al., 2020*; *Mohammadi et al., 2021*; *Bilgin et al., 2022*; *Shamuyarira et al., 2023*; *Yaseen et al., 2024*). When Table S5 is examined, and many combinations can be selected as promising for the trait. It would be more accurate to make a selection by evaluating together with the heterosis and heterobelthiosis values presented in Tables S6 and S7. Because high heterosis and heterobelthiosis values in the $F_1$ generation indicate a performance exceeding the parents, as well as the presence of heterosis-fixing transgressions in the $F_2$ generation (*Mackay et al., 2021*).

Inbreeding depression is stated as the opposite of heterosis, meaning that the fitness of the offspring resulting from inbreeding is reduced (*Merrick et al., 2023*). The presence of inbreeding depression is evidence of directional dominance. Positive values are considered to indicate that the combination has lost hybrid power (*Busa et al., 2022*). The multiplicity of negative values for the traits examined (Table S8) may indicate that the hybrid vigour of the combinations continues, in a way that transgressive lines are selected from the population.

Since it is a tiring and difficult process to get stuck on numbers when choosing the one that suits you among many combinations, breeders are looking for methods that will make this easier and simpler. For this purpose, the regulation of germplasm with heterotic groups makes the breeding program effective. The selection of representative genotypes from each subgroup, identified through geographic origin, morphological data, pedigree information, or molecular markers, facilitates the evaluation of heterotic groups (*Minnu, 2020*).

In this way, crosses between these genotypes are evaluated by selecting the heterotic group based on hybrid performance or components, *i.e.,* heterosis and high observation values in field experiments. Different methods have been developed to form heterotic
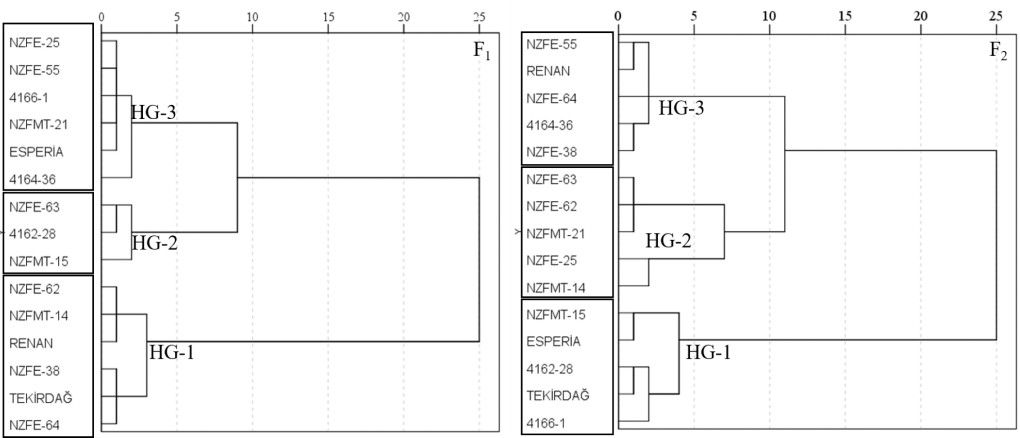

**Figure 1** Dendogram of fifteen parents calculated based on observation means and general combining ability using Euclidean distance matrix.

groups. The determination of the appropriate method will depend on the genetic make-up of the genotypes in the heterotic group. A number of factors have been proposed as criteria for choosing the heterotic group based on SCA, including a high hybrid population mean performance, a high genetic variance, the strong transferability of parental better performance to offspring, and a higher variance of SCA than GCA (*Kutlu & Sirel, 2019*; *Singh et al., 2020*; *Saritha, Umarani & Sridevi, 2022*).

## Heterotic grouping based on combining ability and genotype performance

In this study, the parents were evaluated based on GCA to create a heterotic group, and an assessment was made within the groups created based on SCA. For the heterotic groups created for the parents, the means of yield, GCA and the mean values of the hybrid vigour in the hybrid series entered by the parents were used. As seen in Fig. 1, three heterotic groups were formed in both generations. While NZFE-55 and 4164-36 were in the same heterotic group in both generations, the others were in different groups. The reason why the parents are in different groups in the generations may be the changing environmental conditions and the GCA and hybrid vigour values obtained due to the changing gene distribution of their offspring. The variations presence of inter-group and intra-group indicated that available genotypes were convenient parents to enhance the examined traits. Inter-group variation is between 234.78 (HG2-HG3) and 349.79 (HG1-HG2). There was a maximum intra-group variation for the HG1 and the HG2 followed by it (Fig. 2). It has been reported that good combinations can be obtained from crosses between heterotic group members with large intra-group and inter-group distances (*Gupta et al., 2019*; *Kutlu & Sirel, 2019*; *Paril et al., 2024*).

The tallest plants were collected in HG1 in $F_1$ and HG2 in $F_2$ (Table 2). The mean performances of hybrid series with group member parents (mean-S) in these groups and GCA value were also highest. Heterosis and heterobelthiosis values were moderately high. The group with the highest mean value for spike length also had higher GCA, mean-S and

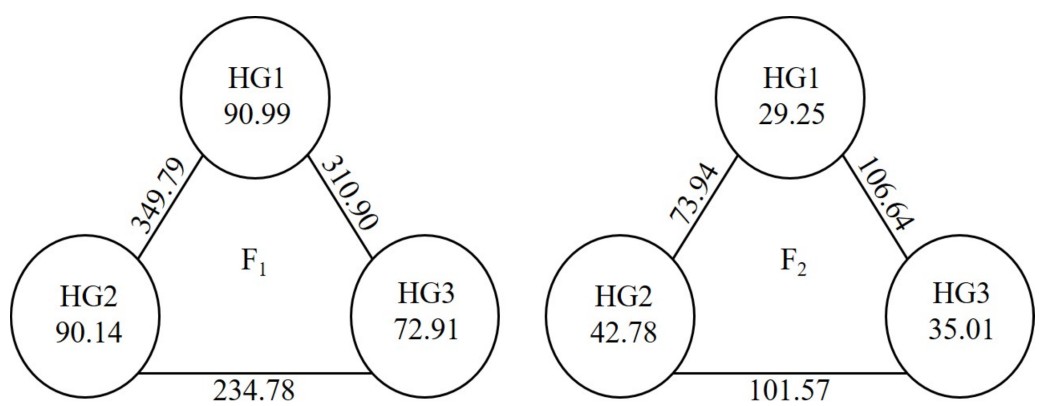

**Figure 2 Cluster diagram showing inter-group and intra-group distances of parental heterotic group.**

heterosis values. Unlike plant height and spike length, in other traits, the group with the highest mean value in terms of HG, GCA, mean-S, Ht, and Hb may not be the highest-value group.

Considering the mean values of $F_1$, the highest values were observed in HG2 for TGW, and in HG1 for GNS, GWS, HI, and GY. As for $F_2$, the highest values were observed in HG3 for all traits except PH, SL and GY. The highest GY was obtained from HG1 (Table 2). In terms of hybrid series performance (mean-S), in $F_1$, the HG2 stood out for all traits, while in $F_2$, HG3 for the TGW and HI, also HG2 for the rest were superior. By examining GCA, Ht and Hb, it is obvious that choosing heterotic groups with positive values would be an appropriate choice.

Bi-plot presents the best input to select effective component created HG such as GCA, Ht, and Hb. In addition, provides information for the accuracy of parents' heterotic groups. When Fig. 3 is examined, it is seen that mean values, Ht and Hb contribute more to the variation required in the formation of heterotic groups than GCA. In $F_1$ with the highest of these values, genotypes 4162-28, NZFE-62, NZFE-63, NZFE-64, NZFMT-15, and Renan can be selected for suitable crossbreeding. In addition, the same parameters as in $F_1$ contribute the most to the variance and the genotypes NZFE-25, NZFE-62, NZFE-63, NZFMT-14, NZFMT-15, and NZFMT-21 can be selected in $F_2$ generation. The formation of heterotic groups in the parents and the changes observed in the genotypes to be selected can be attributed to the environmental conditions. Therefore, it can be recommended that the parents (NZFE-62, NZFE-63 and NZFMT-15) selected in both experimental years (generations) should be particularly emphasised and included in future breeding studies, considering their stability.

The hybrids were classified into three HG similar to their parents in both generations according to the yield-SCA data (Fig. 4). HG1 had nineteen, HG2 had seven, and HG3 had ten members in $F_1$, while HG1 had four, HG2 had fourteen, and HG3 had eighteen members in $F_2$. 4166-1/Tekirdağ and NZFE-55/Renan in the HG1 in $F_2$ were in HG3 in $F_1$, while NZFE-64/Tekirdağ and NZFE-38/Renan joined them from different groups. Twenty-seven of the hybrids took part in the together the HGs in both generations. All

**Table 2** Means of observation, general combining ability, heterosis, heterobeltiosis and inbreeding depression values in parental heterotic groups for examined traits.

| | | F₁ | | | F₂ | | |
| | | HG-1 | HG-2 | HG-3 | HG-1 | HG-2 | HG-3 |
|---|---|---|---|---|---|---|---|
| Plant height (cm) | Mean | 82.75 | 72.87 | 74.35 | 87.26 | 94.84 | 81.60 |
| | GCA | 2.92 | 1.39 | −3.61 | 0.89 | 3.14 | −4.03 |
| | Mean-S | 84.71 | 83.19 | 78.19 | 90.73 | 92.96 | 85.80 |
| | HT | 3.89 | 6.64 | 0.08 | 3.18 | 1.18 | 0.62 |
| | HB | −0.71 | −1.77 | −7.12 | −1.18 | −2.49 | −5.72 |
| | ID | | | | −12.63 | −10.21 | −7.26 |
| Spike length (cm) | Mean | 12.03 | 13.00 | 11.05 | 11.30 | 11.42 | 10.34 |
| | GCA | 0.06 | 0.49 | −0.30 | 0.05 | 0.08 | −0.13 |
| | Mean-S | 11.82 | 12.31 | 11.50 | 10.99 | 11.00 | 10.81 |
| | HT | 2.05 | 4.44 | 4.03 | −1.57 | −1.92 | 0.61 |
| | HB | −3.54 | −4.48 | −2.51 | 0.00 | 0.00 | −0.26 |
| | ID | | | | 4.00 | 7.71 | 8.49 |
| Grain number per spike (number) | Mean | 59.38 | 49.10 | 52.30 | 61.34 | 63.62 | 63.86 |
| | GCA | 1.70 | 2.73 | −3.07 | −0.71 | 1.58 | −0.87 |
| | Mean-S | 57.66 | 58.69 | 52.89 | 61.09 | 63.38 | 60.94 |
| | HT | 0.08 | 7.06 | −4.26 | −3.90 | −2.99 | −4.89 |
| | HB | −5.79 | −2.64 | −10.51 | −8.44 | −5.62 | −9.47 |
| | ID | | | | −14.69 | −7.18 | −18.02 |
| Grain weight per spike (g) | Mean | 1.96 | 1.39 | 1.65 | 2.66 | 2.71 | 2.76 |
| | GCA | 0.04 | 0.19 | −0.14 | 0.00 | −0.12 | −0.04 |
| | Mean-S | 2.18 | 2.33 | 2.00 | 3.15 | 3.18 | 3.11 |
| | HT | 18.34 | 37.66 | 13.12 | 11.61 | 9.75 | 10.18 |
| | HB | 6.59 | 18.18 | 1.64 | 2.58 | 3.22 | 0.23 |
| | ID | | | | −56.96 | −46.71 | −54.60 |
| Thousand kernel weight (g) | Mean | 30.89 | 30.77 | 31.96 | 41.29 | 41.98 | 42.94 |
| | GCA | −0.02 | 0.98 | −0.47 | 0.16 | −0.39 | 0.23 |
| | Mean-S | 35.35 | 36.36 | 34.90 | 46.29 | 45.73 | 46.36 |
| | HT | 12.51 | 12.56 | 7.41 | 9.35 | 6.69 | 6.78 |
| | HB | 5.15 | 7.87 | 2.44 | 5.42 | 2.17 | 2.98 |
| | ID | | | | −28.63 | −36.63 | −30.71 |
| Harvest index (%) | Mean | 37.98 | 30.48 | 34.79 | 49.83 | 49.42 | 51.13 |
| | GCA | 1.12 | 2.53 | −2.39 | −0.36 | −0.24 | 0.60 |
| | Mean-S | 40.44 | 41.85 | 36.94 | 49.25 | 49.37 | 50.21 |
| | HT | 9.49 | 23.10 | 3.39 | −1.88 | −1.37 | −0.90 |
| | HB | 4.46 | 11.18 | −3.74 | −3.39 | −2.89 | −2.52 |
| | ID | | | | −31.82 | −22.53 | −30.90 |
| Grain yield (t ha⁻¹) | Mean | 6.72 | 3.32 | 3.98 | 6.38 | 6.22 | 5.35 |
| | GCA | 42.53 | 121.64 | −103.35 | −31.56 | 40.11 | −8.55 |
| | Mean-S | 5.87 | 6.66 | 4.41 | 6.12 | 6.84 | 6.35 |
| | HT | −1.11 | 47.56 | −6.98 | 1.63 | 15.25 | 14.75 |
| | HB | −14.68 | 16.98 | −21.50 | −4.88 | 8.03 | 8.57 |
| | ID | | | | −31.73 | −37.76 | −36.46 |

**Notes.**

HG, Heterotic group; Mean, True mean values of parents; GCA, General combining ability; Mean-S, Mean value of each parent in the crossbreeding series; HT, heterosis value of each parent in the crossbreeding series; HB, heterobeltiosis value of each parent in the crossbreeding series; ID, inbreeing depression value of each parent in the crossbreeding series.

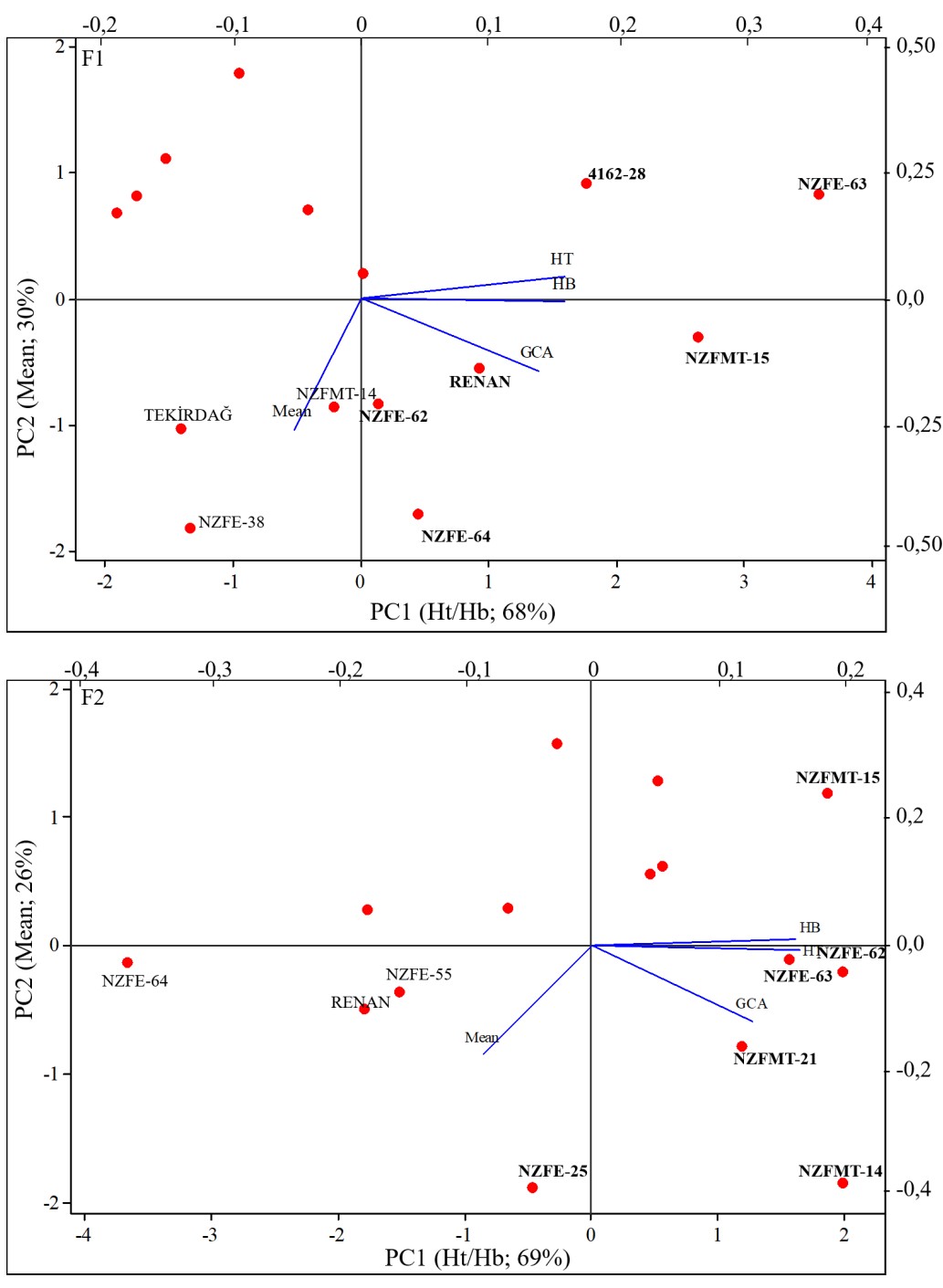

**Figure 3** Bi-plot that created from heterotic groups for selection best parents.

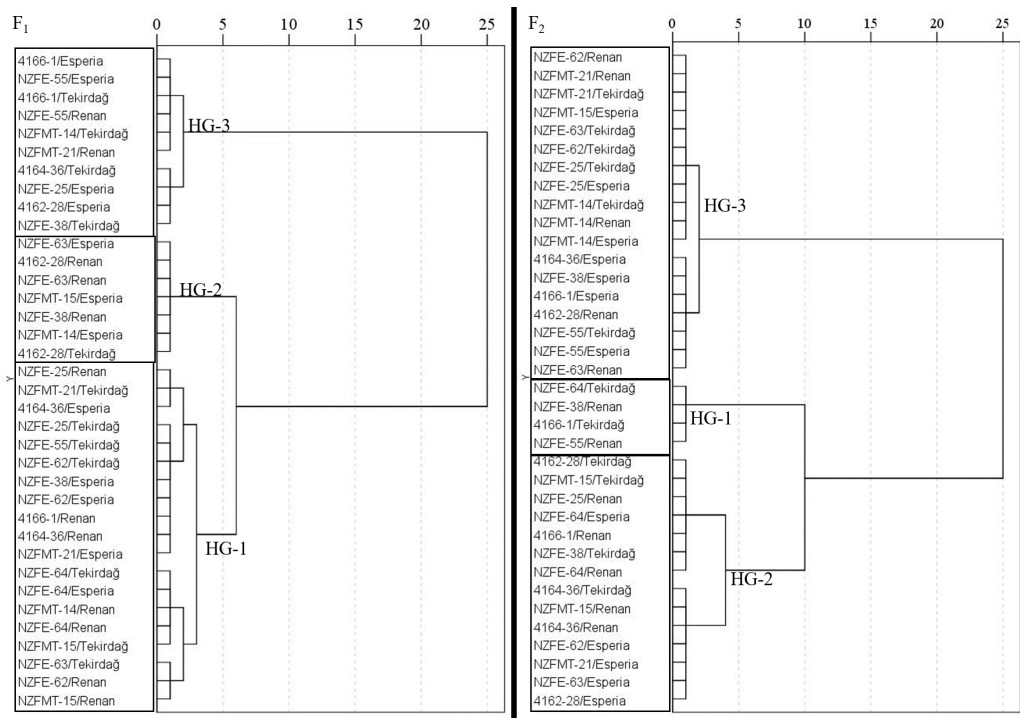

**Figure 4** Dendrogram showing heterotic grouping based on the specific combining ability of hybrids.

combinations of NZFE-62 and NZFE-64 parents were included in the same HG in $F_1$. In $F_2$, all combinations of NZFMT-14 were in the same group. For these combinations, it is thought that the maternal effect of their characteristics may be greater due to the female parent.

While the intra-group distance of HG1 and HG3 was greater in $F_1$, it was the maximum in HG2 in $F_2$ (Fig. 5). While the distances of inter-groups were the highest between HG2 and HG3 in $F_1$, the greatest distance was between HG1 and HG3 in $F_2$. The values in $F_1$ were higher than those in $F_2$.

Hybridization between members of the different groups with the farthest HG in $F_1$ and $F_2$ can result in superior combinations in terms of high variation and desired characteristics.

For each characteristic, the means, SCA, Ht, Hb, and ID values of the hybrids were assessed in HGs (Table 3). The HG-3 group in the $F_1$ generation had the lowest mean value for plant height, yet their SCA value was negative and their Ht-Hb values were lower than those of HG-1 and HG-2. The hybrids from the $F_2$ generation with the lowest values were grouped together in HG-2. As for the shortness, the hybrids in these groupings can be regarded as promising. When HGs was examined in terms of spike length, it was seen that genotypes with the highest values were found in HG-2 in both generations. SCA, Ht and Hb values calculated for spike length were not high, and positive ID in $F_2$ indicates that the effects of the gene that increases spike length have decreased. However, spike length is a secondary yield component and can be compensated for by improving the grain number and weight per spike, which are the primary yield components (*Gaju et al.,*

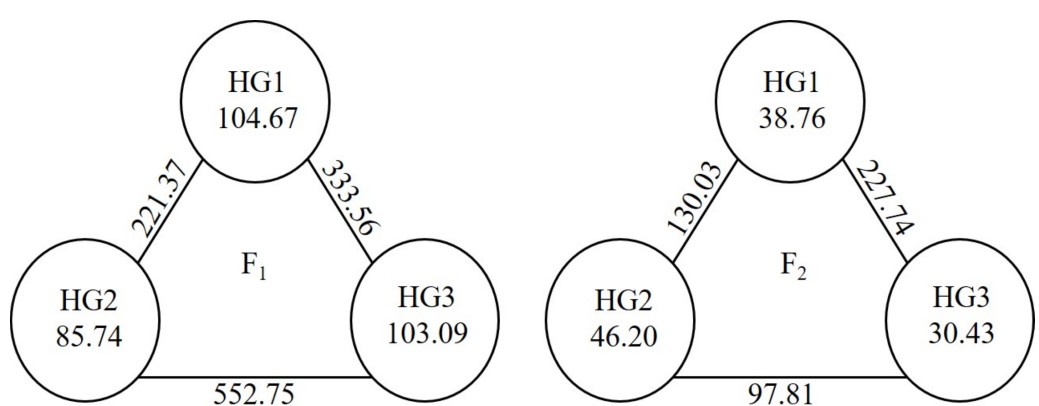

**Figure 5** Cluster diagram showing inter-group and intra-group distances of heterotic group of hybrids.

*2009*). In particular, positive SCA, Ht and Hb values in the grain weight of the spike in $F_2$ and negative ID indicate the presence of transgressive segregations and the presence of promising combinations for grain weight per spike in these groups. The mean values of the HGs are also close to each other, therefore, considering the combinations with the highest values, positive SCA, Ht and Hb values in the group and bringing these traits from $F_1$ may increase the probability of success. High thousand grain weight in wheat is a desired breeding goal (*Zhang, Zheng & He, 2021*). Combinations with the highest mean, SCA, Ht and Hb values in HG2 in $F_1$ predominate. Among these, 4162-28/Tekirdağ and NZFE-63/Esperia took place in $F_2$ in HG2 with the highest value. Since HG2 also has positive SCA, Ht and Hb values, the mentioned combinations should be followed for high thousand grain weight. Although some combinations do not have the highest value, they can come to the fore with high SCA, Ht and Hb values. Such combinations may be superior in terms of their ability to transfer the trait-enhancing abilities they have to future generations or when they become parents. The HGs with high mean value, positive SCA, Ht and Hb value and negative ID value can be selected to improve the studied traits. Combinations that are members of these groups should be followed. Positive SCA indicates the positive development of the related trait, while positive Ht and Hb and negative ID indicate the presence of fixed heterosis due to transgressive segregation (*Mackay et al., 2021*).

The best combinations for harvest index and grain yield were in HG2 in $F_1$. The HG was with the highest mean, SCA, Ht and Hb values. The harvest index in HGs demonstrated a significant negative heterosis, followed by a negative inbreeding depression. This suggests the occurrence of additive gene effects, which would result in the appearance of transgressive segregants in the $F_2$ and subsequent generations. Consequently, it is possible to isolate superior-yielding genotypes from the segregating population by isolating desirable segregants.The highest mean yield, SCA, Ht and Hb values were observed in HG3 in $F_2$, which HG contains most members of HG2 in $F_1$. Combinations of 4162-28/Renan,

**Table 3** Means of observation, specific combining ability, heterosis, heterobeltiosis and inbreeding depression values in heterotic groups of hybrids for examined traits.

| | | $F_1$ | | | $F_2$ | | |
|---|---|---|---|---|---|---|---|
| | | HG-1 | HG-2 | HG-3 | HG-1 | HG-2 | HG-3 |
| Plant height (cm) | Mean | 83.74 | 81.60 | 78.25 | 89.28 | 88.42 | 91.04 |
| | SCA | 0.67 | −1.34 | −0.34 | −0.39 | 0.23 | −0.09 |
| | HT | 2.93 | 4.14 | 1.55 | 1.67 | 1.40 | 1.98 |
| | HB | −2.63 | −4.27 | −4.73 | −2.01 | −4.34 | −2.55 |
| | ID | | | | −10.93 | −9.31 | −10.39 |
| Spike length (cm) | Mean | 11.84 | 12.31 | 11.33 | 10.98 | 11.01 | 10.86 |
| | SCA | 0.00 | 0.21 | −0.16 | 0.17 | 0.08 | −0.10 |
| | HT | 2.49 | 5.11 | 2.50 | −2.17 | 0.49 | −1.50 |
| | HB | −3.98 | −2.97 | −2.81 | −6.51 | −3.79 | −5.72 |
| | ID | | | | 7.29 | 5.68 | 7.43 |
| Grain number per spike (number) | Mean | 57.89 | 57.61 | 51.12 | 58.90 | 62.05 | 62.26 |
| | SCA | 1.02 | 0.08 | −1.99 | −0.58 | 0.93 | −0.59 |
| | HT | 0.41 | 6.71 | −7.72 | −5.00 | −3.79 | −4.10 |
| | HB | −5.36 | −0.58 | −15.61 | −10.36 | −7.57 | −7.50 |
| | ID | | | | −9.12 | −19.13 | −9.68 |
| Grain weight per spike (g) | Mean | 2.25 | 2.24 | 1.86 | 3.14 | 3.18 | 3.12 |
| | SCA | 0.08 | 0.00 | −0.15 | 0.03 | 0.06 | −0.05 |
| | HT | 22.55 | 30.54 | 4.22 | 14.80 | 11.39 | 8.95 |
| | HB | 11.09 | 12.92 | −7.32 | 2.10 | 3.20 | 1.32 |
| | ID | | | | −58.63 | −60.78 | −45.21 |
| Thousand kernel weight (g) | Mean | 34.99 | 37.52 | 34.60 | 46.46 | 46.74 | 45.57 |
| | SCA | −0.19 | 1.15 | −0.44 | −0.12 | 0.50 | −0.36 |
| | HT | 9.48 | 15.61 | 7.15 | 10.36 | 9.08 | 6.70 |
| | HB | 3.89 | 10.03 | 2.00 | 7.98 | 4.94 | 2.19 |
| | ID | | | | −22.88 | −31.53 | −34.37 |
| Harvest index (%) | Mean | 40.44 | 44.23 | 33.77 | 48.67 | 49.97 | 49.53 |
| | SCA | 0.96 | 2.49 | −3.57 | −0.38 | 0.43 | −0.25 |
| | HT | 11.42 | 27.66 | −7.01 | −2.64 | −1.15 | −1.40 |
| | HB | 5.02 | 16.76 | −13.07 | −4.17 | −2.29 | −3.20 |
| | ID | | | | −26.70 | −30.72 | −27.01 |
| Grain yield (t ha$^{-1}$) | Mean | 5.84 | 7.72 | 3.10 | 5.08 | 6.15 | 6.96 |
| | SCA | 31.71 | 126.40 | −148.74 | −86.22 | −18.14 | 33.27 |
| | HT | 8.09 | 64.97 | −37.94 | −16.65 | 6.29 | 19.43 |
| | HB | −8.18 | 32.62 | −45.84 | −19.42 | −0.27 | 11.65 |
| | ID | | | | −9.72 | −28.03 | −46.68 |

**Notes.**

HG, Heterotic group; Mean, Mean values of hybrids; SCA, Specific combining ability; HT, heterosis value; HB, heterobeltiosis value; ID, inbreeing depression value.

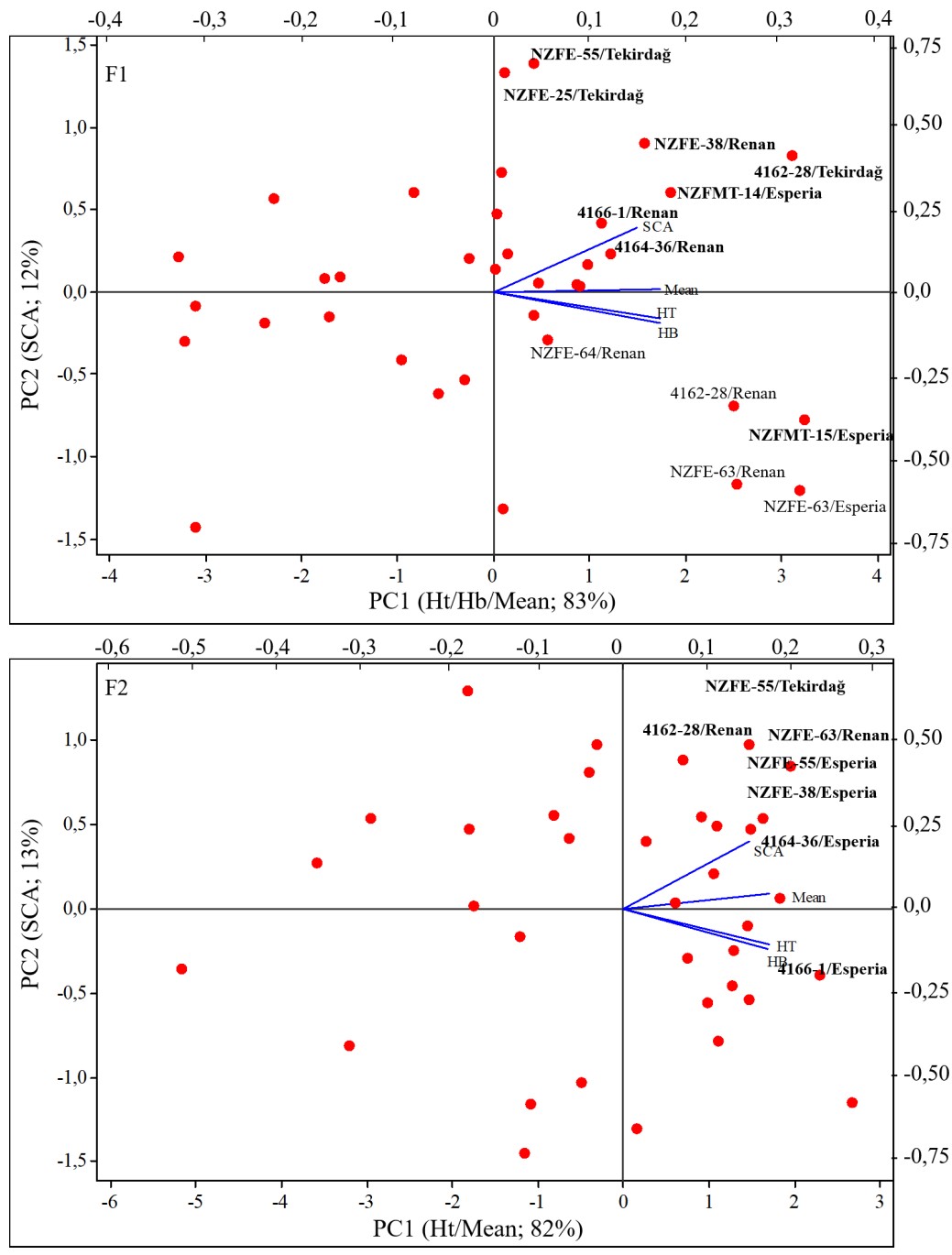

**Figure 6** **Bi-plot that created from heterotic groups for selection best combinations.**

NZFE-63/Renan, NZFMT-15/Esperia, and NZFMT-14/Esperia in high-value HGs in both generations should be followed to develop high-yielding lines.

The bi-plot to confirm heterotic groups for hybrid combinations is shown in Fig. 6. It is seen that the selected variables are effective in the formation of heterotic groups and it

is appropriate to form heterotic groups based on SCA. Selected combinations in heterotic groups were plotted on the graph. NZFE-55/Tekirdağ and 4162-28/Renan, NZFE-63/Renan combinations selected from high-value HGs and marked on the biplot graph should be considered.

## CONCLUSIONS

The parents of the mentioned combinations were also selected from the GCA-based heterotic groups. In this case, the current parent's choices are already appropriate. For future studies, new and different combinations can be created by considering these heterotic groups. Heterotic grouping can be used effectively in breeding programs to select both proper parental and superior combinations. It may be necessary to conduct molecular marker analysis in order to corroborate the genotypes' genetic separation. Nevertheless, this will merely serve as a supplementary outcome. Therefore, gained results pointed out that a more useful tool the heterotic groups based on SCA than ones based on GCA for future breeding studies. Since all genotypes within the heterotic groups created in this way are selected, it will be easier to retain those with superior characteristics for a longer period of time in breeding programs. At the same time, creating heterotic groups as reported here will enable the most accurate selection to be made in a shorter time and at less cost.

## ACKNOWLEDGEMENTS

A part of this study includes Birol Deviren's master's thesis.

### Funding
The authors received no funding for this work.

### Competing Interests
Imren Kutlu is an Academic Editor for PeerJ.

### Author Contributions
- Birol Deviren conceived and designed the experiments, performed the experiments, prepared figures and/or tables, and approved the final draft.
- Oguz Bilgin conceived and designed the experiments, performed the experiments, authored or reviewed drafts of the article, and approved the final draft.
- Imren Kutlu conceived and designed the experiments, analyzed the data, prepared figures and/or tables, authored or reviewed drafts of the article, and approved the final draft.

### Data Availability
 The all raw measurements are available in the Supplementary File.

## Supplemental Information

Supplemental information for this article can be found online at http://dx.doi.org/10.7717/peerj.18136#supplemental-information.

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
