# Peer review of "Heterotic grouping of wheat hybrids based on general and specific combining ability from line × tester analysis"

_PeerJ, doi:10.7717/peerj.18136_

## Round 0.1 · original submission · Major Revisions

· Academic Editor

Major Revisions

The authors need to include all suggestions given by reviewers.

Reviewer 1 ·

Basic reporting

Dear Author,
I have carefully reviewed the article “Heterotic grouping based on general and speciûc combining ability in line × tester wheat hybrids” where I find the study a good research work towards crop improvement programme. Though studies have been reported till date regarding wheat breads but one study was found earlier mentioned in the comment section which is close to this current study which needs an extensive explanation. Moreover, some major mistakes are found which are mentioned below therefore my suggestion will be that to go thoroughly through the whole manuscript and rectify as needed.
In abstract part
1. Keywords are missing in the abstract part
In introduction part
1. In Line 36 is not well written. The importance of Wheat crop is needed worldwide where turkey is also included. So it is not required to mention separately. kindly rewrite it
2. In line 39, Here the report is from 2015, Can you add a recent report?
3. In 51-52, Add a reference here. If a certain informative line is provided it is good to add a reference
4. Lack of reference in this paragraph. From line 49-57, there is only one reference for the whole paragraph. It is good to cite more references for any informative lines. kindly add
5. In line 93, Can you explain what is the difference between current study and this study Kutlu, İ., & Sırel, Z. (2019). Using line× tester method and heterotic grouping to select high yielding genotypes of bread wheat (Triticum aestivum L.). ?
6. In the end part of Introduction, Mention elaborately about this current study how it is novel from other studies?

Experimental design

In materials and method
1. In line 122-124, As in the discussion part the author have added abbreviation of these traits, and readers may not understand, Kindly add abbreviation of these traits in materials and methods part

Validity of the findings

In result part
1. In line 203-204, Reframe the line, usually we dont write "as you can see" in a scientific paper
2. In line 216, Cite few references
3. line 245-246 is not clear, rewrite
Conclusion
1. write little more about importance of the current study outcomes in future breeding program
2. the line 314-315 is not clear, rewrite it

Additional comments

References
1. References are not according to the format. Kindly go through the whole references and rectify it accordingly.

Reviewer 2 ·

Basic reporting

The study titled “Heterotic grouping based on general and specific combining ability in line x tester wheat hybrids” focuses on analyzing yield, genetic diversity, and the development of heterotic groups in bread wheat using line x tester wheat populations. The authors evaluate the F1 and F2 generations of 51 genotypes, comprising 36 combinations between 12 lines and 3 testers. The general and specific combining abilities, heterosis, heterobeltiosis, and inbreeding depression is calculated as well. With the identification of sufficient genetic variation in the population, enabling the formation of heterotic groups for selecting superior genotypes based on various criteria. All these results suggest that the heterotic groups formed mainly based on specific combing abilities may be more useful for breeding studies. Overall, this research provides insights into yield and genetic diversity in line x tester wheat populations, with implications for breeding programs and the selection of superior genotypes. The manuscript is written in a clear and logical manner, however some minor problems need revision.

Experimental design

All experiments are designed reasonable.

Validity of the findings

Conclusions are well stated.

Additional comments

I suggest the author check and correct the citation format through the manuscript main text to improve consistency.
I suggest improving the resolution of Figure 1 and Figure 4.
Does the author have the PC score figure?

---

## Round 0.2 · Minor Revisions

· Academic Editor

Minor Revisions

The results data regarding genotype-wise values of GCA, SCA, heterosis, heterobeltiosis and inbreeding depression are missing in Result section.

Biplot figures are not clear to read. These figure do not show the axis (X, Y) wise PCs

Heterotic grouping of lines should be based on their positive SCA values of grain yield. To create heterotic groups of lines, the researchers should use those testers with known heterotic groups.

It is not clearly shown that which genotypes belong to which heterotic groups. This information in tabular form is lacking.

Reviewer 1 ·

Basic reporting

Dear authors,
I have carefully re-reviewed the article entitled "Heterotic grouping based on general and specific
2 combining ability in line × tester wheat hybrids" where all the mistakes has been rectified accordingly and now the MS does 'not required any changes.

Experimental design

NA

Validity of the findings

NA

Additional comments

NA

Reviewer 2 ·

Basic reporting

Thank you for working on the revision. The manuscript has been improved upon reviewers' request.

Experimental design

NA

Validity of the findings

Figure 3, are the two grey circles drawn by hand? The figure needs to be edited to be more professional. Also Figure 3 was given very little information in the main text, it's hard for audience to understand especially when there is no figure legend.

Additional comments

NA

---

## Round 0.3 · accepted · Accept

· Academic Editor

Accept

I confirm that the article is now Acceptable. I simply request that you correct the title as per: "Heterotic grouping of wheat hybrids based on general and specific combining ability from line × tester analysis". This can be done while in production

Reviewer 1 ·

Basic reporting

Dear Author,
The MS has been revised and rectified accordingly, and it is in right format to get accepted.

Experimental design

NA

Validity of the findings

NA

Reviewer 2 ·

Basic reporting

NA

Experimental design

NA

Validity of the findings

NA

Additional comments

Given the thorough incorporation of point-by-point suggestions throughout the manuscript, I recommend its acceptance for publication. Thanks!